# Single-molecule junction spontaneously restored by DNA zipper

Takanori Harashima[1], Shintaro Fujii [1], Yuki Jono[1], Tsuyoshi Terakawa [2], Noriyuki Kurita[3], Satoshi Kaneko [1], Manabu Kiguchi[1] & Tomoaki Nishino [1✉]

The electrical properties of DNA have been extensively investigated within the field of molecular electronics. Previous studies on this topic primarily focused on the transport phenomena in the static structure at thermodynamic equilibria. Consequently, the properties of higher-order structures of DNA and their structural changes associated with the design of single-molecule electronic devices have not been fully studied so far. This stems from the limitation that only extremely short DNA is available for electrical measurements, since the single-molecule conductance decreases sharply with the increase in the molecular length. Here, we report a DNA zipper configuration to form a single-molecule junction. The duplex is accommodated in a nanogap between metal electrodes in a configuration where the duplex is perpendicular to the nanogap axis. Electrical measurements reveal that the single-molecule junction of the 90-mer DNA zipper exhibits high conductance due to the delocalized π system. Moreover, we find an attractive self-restoring capability that the single-molecule junction can be repeatedly formed without full structural breakdown even after electrical failure. The DNA zipping strategy presented here provides a basis for novel designs of single-molecule junctions.

[1] Department of Chemistry, School of Science, Tokyo Institute of Technology, 2-12-1 W4-11 Ookayama, Meguro-ku, Tokyo 152-8551, Japan. [2] Department of Biophysics, Graduate School of Science, Kyoto University, Kitashirakawa-Oiwakecho, Sakyo, Kyoto 606-8502, Japan. [3] Department of Computer Science and Engineering, Toyohashi University of Technology, Tempaku-cho, Toyohashi 441-8580, Japan. ✉email: tnishino@chem.titech.ac.jp

The structural, physical, and chemical properties of DNA at the nanoscale have attracted attention to the prospect of using DNA as a building block in nanoscience. In the field of molecular electronics, electron transport through a single DNA molecule has been extensively investigated[1–5]. These studies exploit single-molecule junctions, in which a DNA molecule bridges a nanogap between the metal electrodes. For example, it was reported that A-form and B-form DNAs differ in the single-molecule conductance by an order of magnitude[6]. Many studies have revealed that there is a change in the electron transport properties of DNA after ligand binding[7–9]. These include an intercalator whose perturbation of base stacking leads to switching or rectifying properties of DNA[10–12]. Moreover, a three-terminal DNA connected by a guanine quadruplex has been exploited to construct a single-molecule junction, and it was demonstrated that the intricate junction structure shows excellent functionality for a charge splitter[13]. Significant advances have been made in understanding the transport phenomena through a single DNA molecule, as exemplified above. However, these previous studies mainly focused on the transport phenomena in static DNA structure at thermodynamic equilibria.

Recently, rational control of DNA structures at the single-molecule level has been achieved by modern techniques and emerging technologies. Capturing desired conformation and separation of DNA are achieved by elaborate flow in nanofluidic systems[14–17]. Sequencing analysis of DNA at the single-base resolution can be performed with nanopore devices during the constrained passage[18–21]. Also, precise control of DNA structures by atomic force microscopy (AFM) has enabled accurate mapping of DNA unzipping dynamics and a free-energy landscape[22–24]. This combination of electrical measurements and structural modulation of DNA could lead to the realization of the sophisticated functionality of electronic devices based on a single DNA molecule. For example, we expect that electron transport through DNA under the deliberate control of its structure paves the way for DNA electronic devices with functional controllability in a dynamic manner.

Here, we report the investigation of electron transport through the single-molecule junction of a zipper DNA that orthogonally clamps a metal nanogap (see Fig. 1a). The present DNA single-molecule junction differs from conventional ones in the DNA configuration; the present and conventional junctions contain a DNA molecule oriented in perpendicular and parallel directions to the axis of the nanogap, respectively. The DNA single-molecule junction with the present zipper configuration exhibited high single-molecule conductance that cannot be achieved using DNA junctions with the conventional configuration. The unzipping dynamics of the molecular junction were characterized by tunnelling currents in break-junction (BJ) experiments, based on scanning tunnelling microscopy (STM). The STM measurement and the molecular dynamics (MD) simulations of the unzipping dynamics revealed that the DNA junction with zipper configuration enables spontaneous restoration of the molecular junction after its electrical failure and thereby improves the reproducibility of the junction formation. Our study demonstrated that a DNA dynamic structural change could be applied to a single-molecule junction by using the zipper configuration. The findings pave way for novel functionality and superior properties of nanoscale electronic devices[22,25].

## Results and discussion

**Single-molecule conductance.** First, STM-BJ experiments were performed to measure conductance of the single-molecule junction of DNA with a zipper configuration. Either 90-mer or 10-mer DNA was employed as a sample molecule. One strand of the sample was functionalized at the 3′ end with a thiol linker, while the same linker was introduced at the 5′ end of another strand (Fig. 1a). These ends were tethered to the STM tip or Au(111) surface. The STM tip was repeatedly brought in and out of contact with the Au(111) surface modified with the DNA duplex. The conductance was monitored as a function of the tip displacement during the retraction process (Fig. 1b). The resulting two-dimensional (2D) histogram, where thousands of conductance–displacement ($G$–$z$) traces were overlaid, clearly

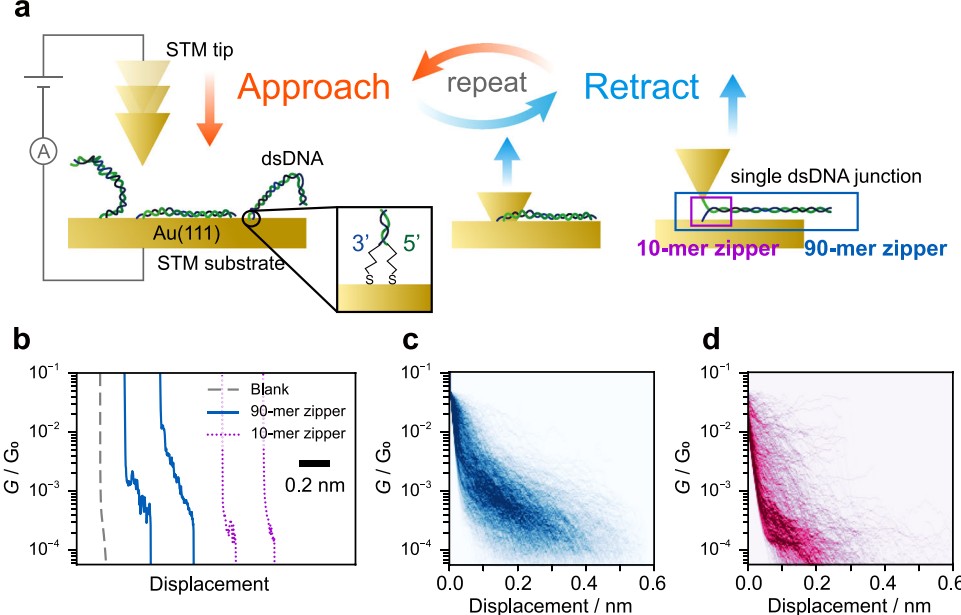

**Fig. 1 Single-molecule junction of DNA zipper. a** Schematic illustration of the scanning tunnelling microscopy–break-junction (STM-BJ) measurements. **b** Representative conductance traces for unmodified Au(111) substrate (graydashed lines), and 90-mer and 10-mer DNAs (blue solid lines and purple dotted lines, respectively). **c** and **d** 2D histograms of the conductance–displacement ($G$–$z$) traces for 90-mer and 10-mer DNAs, respectively. The origin of the displacement was set at the point where the conductance decreased below 50 m$G_0$. 2219 and 2635 traces were analyzed for histograms in **c** and **d**, respectively. Tip velocity, 31 nm/s; bias voltage, 20 mV.

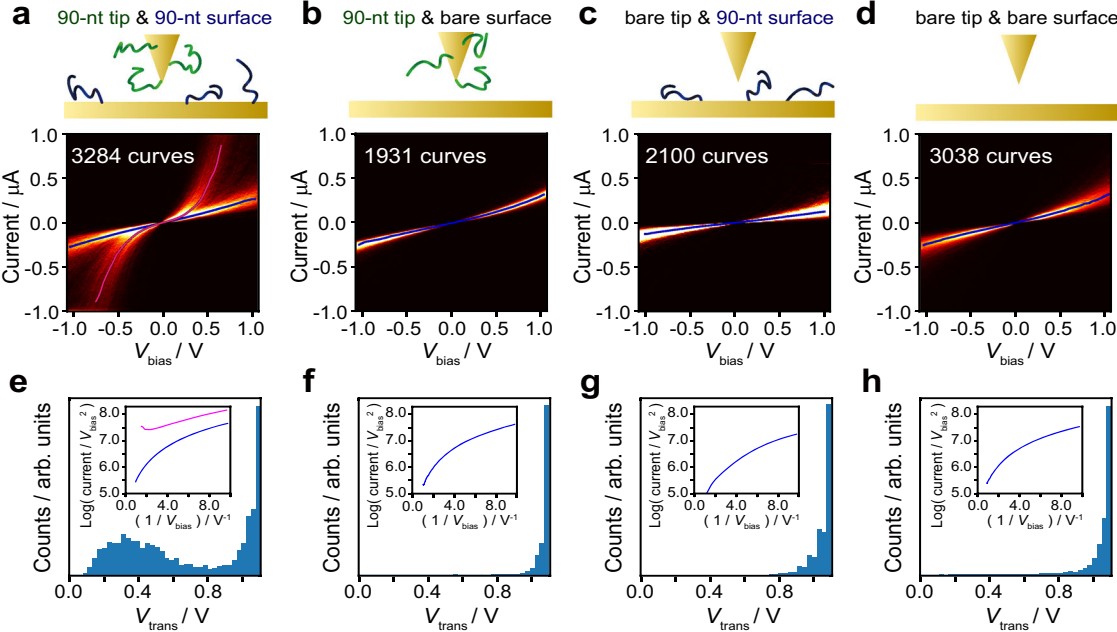

**Fig. 2 Current–voltage (*I–V*) and transition voltage spectra.** *I–V* 2D histograms and $V_{trans}$ histograms obtained with (**a, e**) molecular tip and modified substrate, (**b, f**) molecular tip and bare substrate, (**c, g**) unmodified tip and modified substrate, and (**d, h**) unmodified tip and bare substrate. Bias voltage ($V_{bias}$) was swept from −1.1 V to 1.1 V in 5 ms. Representative transition voltage spectra are shown in insets of **e–h**. Pink and blue spectra correspond to those with low and high $V_{trans}$, respectively.

shows conductance plateaus at 1.9 and 0.15 $mG_0$ for 90-mer and 10-mer DNA, respectively (Fig. 1c, d). The difference in the single-molecule conductance is discussed later. The single plateau and subsequent conductance decay in the traces indicate that these plateaus are attributed to the single-molecule junction that contains DNA. The tunnelling decay constants during and after the plateau ($\beta_1$ and $\beta_2$, respectively) were analysed from each conductance trace, and $\beta_1$ and $\beta_2$ for 90-mer DNA were determined to be 0.27 and 2.0 Å$^{-1}$, respectively (see Supplementary Note 1 for the results of 10-mer DNA and detailed discussion of STM-BJ results). The decay constants are known to depend on the energy gap between the Fermi level of the electrode and that of the molecular orbital for the tunnelling transport[26]. The $\beta_2$ value found here is consistent with that for direct tunnelling between the tip and substrate without the molecular junction (2.2 Å$^{-1}$, Supplementary Note 1). On the other hand, the $\beta_1$ value is smaller than the typical value for alkanedithiol, but similar to the ones for π-conjugated molecules[26], which indicates that the electron transport involves the DNA. The DNA zipper junction transmits electrons in the transverse direction, and it is anticipated that the base pairs, especially those located at the DNA terminal, mediate the electron transport (see Fig. 1a). In this case, the transport properties of the present junction can be compared with those of the single-molecule junction of DNA bases, which have been investigated toward the realization of single-molecule sequencing[27–30]. Indeed, the reduction in the decay constants as observed for $\beta_1$ value in the present experiments was reported for the tunnelling through the DNA base pairs[31,32]. Further STM-BJ experiments showed that the conductance of the molecular junction reflects the DNA sequence (Supplementary Note 2), which further supports that the electron transport is mediated by the DNA.

The present DNA single-molecule junction possesses a different configuration from the conventional one. In the present study, the DNA duplex bears the two linker groups at the same end. Thus, the molecular junction contains DNA in a configuration orthogonal to the axis of the gap between the tip and

substrate (Fig. 1a). In contrast, the linker groups were conventionally located at the opposite ends of the duplex, and the junction accommodates the duplex aligned parallel to the gap axis. Electron transport through the latter conventional junction steeply attenuates with increased DNA length, since the electrons travel through the whole duplex[33]. No such attenuation happens in the present single-molecule junction. Indeed, the conductance of the 90-mer DNA zipper junction is larger than that of the 10-mer zipper junction. We measured the conductance of the zipper junction of a variety of DNA lengths ranging from 10 to 90 base pairs and confirmed that the conductance value increased as the DNA length increased (Supplementary Note 3). This length dependence, together with the self-restoring capability described later, are the advantages of the present configuration.

**Transition voltage spectroscopy.** Next, current–voltage (*I–V*) curves of the single-molecule DNA junction with the zipper configuration were obtained to investigate the electron transport properties in detail. We previously developed a methodology to study electron transport induced by in situ hybridization of a single DNA duplex using an STM molecular tip[7,34]. This technique was also utilized in the present study: an Au STM tip was modified with a single strand of the 90-mer DNA zipper. An Au(111) substrate was separately modified with single-stranded DNA (ssDNA) complementary to the strand on the tip (Fig. 2a). After the molecular tip was brought near to, but never in contact with, the surface, the bias voltage was swept with the tip as the sample distance was held stationary to acquire the *I–V* curves. Figure 2a shows the resulting *I–V* curves, and Fig. 2b–d presents those obtained in the control experiments, i.e., the measurements with the molecular tip and unmodified surface, with the unmodified tip and ssDNA-modified substrate, and with the unmodified tip and unmodified surface, respectively. The *I–V* curves in Fig. 2a clearly exhibited two distributions, i.e., states with high and low conductance. The low-conductance state was common to those found in the control experiments (Fig. 2b–d), indicating that this state stems from the gap devoid of the DNA bridge. The

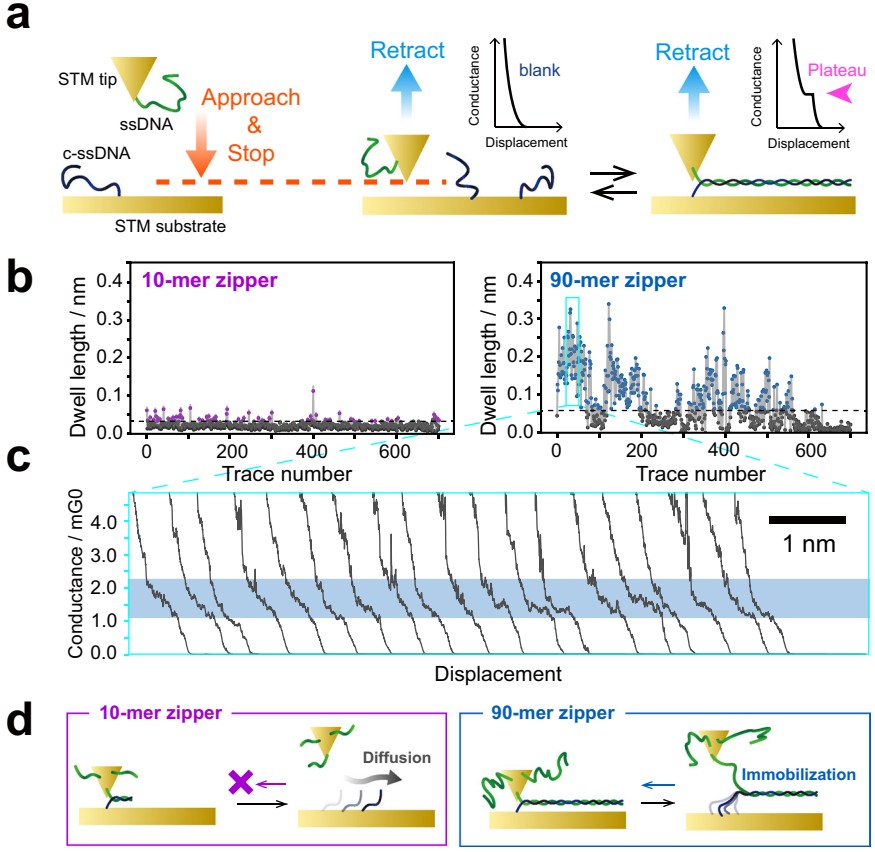

**Fig. 3 Consecutive conductance–displacement (G–z) measurements for molecular junctions of DNA zipper. a** Schematic illustration of in-situ hybridization using STM molecular tip. Initial set-point current, 7.5 nA; pull-up distance, 30 nm; tip velocity, 31 nm/s. **b** Temporal evolution of the dwell length of G–z trace for 10-mer (left) and 90-mer (right) DNA. The dwell length was calculated as the trace length between 0.14 and 0.16 mG$_O$ and between 1.4 and 2.4 mG$_O$ for the 10-mer and 90-mer zipper, respectively. The threshold values for 10-mer and 90-mer DNA were 0.028 and 0.060 nm, respectively, as indicated by dashed lines. **c** Successive G–z traces, extracted from the area indicated by square in **b**. **d** Plausible model for the formation and breakdown of molecular junctions in DNA zipper.

high-conductance state was thus ascribed to the molecular junction of the DNA zipper. The conductance as estimated by the I–V properties agrees with the conductance as determined by the BJ studies (Fig. 1c, d, see Supplementary Note 4), supporting the assignment of a high-conductance state to the DNA junction. It has been reported that ssDNA can adsorb to a metal surface via its bases[35]. However, no state that could be attributed to ssDNA molecular junctions was found in the I–V curves in Fig. 2b, c. STM-BJ study was also conducted with the unmodified tip and the ssDNA-modified substrate, and the conductance histograms without notable peaks were obtained (Supplementary Note 5). These results are most probably due to the significantly decreased conductance of ssDNA as compared to that of dsDNA because of base stacking is less ordered in ssDNA[36]. The conductance of the ssDNA junction would be almost indistinguishable from the conductance of the gap without the dsDNA bridge in the present measurements.

For evaluation of the electronic structure of the molecular junction, the transition voltage ($V_{trans}$) was estimated using I–V curves of the high-conductance state (Fig. 2e–h). It has been known that $V_{trans}$ is proportional to the energy gap between the Fermi level of the electrode (the tip or the substrate) and that of the conduction orbital of the molecule in the junction[37–39]. In transition voltage spectra, $\log(I/V^2)$ is plotted against $V^{-1}$, and the voltage at which the plot reaches a minimum corresponds to $V_{trans}$[37–39]. The mean value of $V_{trans}$ for the molecular junction of the 90-mer DNA zipper was found to be 0.4 V (Fig. 2e). In

our previous work, a $V_{trans}$ value of 0.8 V was found for a single-molecule junction of 8-mer DNA in the conventional configuration. It was further observed that this value decreased to 0.5 V upon binding of an intercalator to the DNA[11]. The $V_{trans}$ value for the molecular junction of the DNA zipper is smaller than in both these above-mentioned cases. This result suggests the decrease of the energy gap by delocalization of π stacking orbitals over its long base pairs, which agrees with previous research that used fragment molecular orbital and density-functional theory calculations[40,41]. We attribute the small $V_{trans}$, that is, the decreased energy gap, to delocalization of the π-orbitals of DNA over its long base pairs. To prove this, molecular orbital calculations were performed based on density-functional theory (Supplementary Note 6). We indeed found that the energy gap between the highest occupied molecular orbital (HOMO) and the lowest unoccupied molecular orbital (LUMO) decreased with the increase in DNA length, in line with previous theoretical studies[40,41]. The larger conductance of the 90-mer DNA junction compared to the 10-mer counterpart (Fig. 1c) is consistent with the length-dependent decrease in the HOMO–LUMO gap. Thus, we conclude that the single-molecule junction of the DNA zipper attains high conductance due to the delocalized π system of the stacked DNA bases near the electrodes. The effect of this electron delocalization could explain the higher conductance of the 90-mer DNA zipper compared with that of the 10-mer counterpart (Fig. 1b), though this behavior merits further investigation.

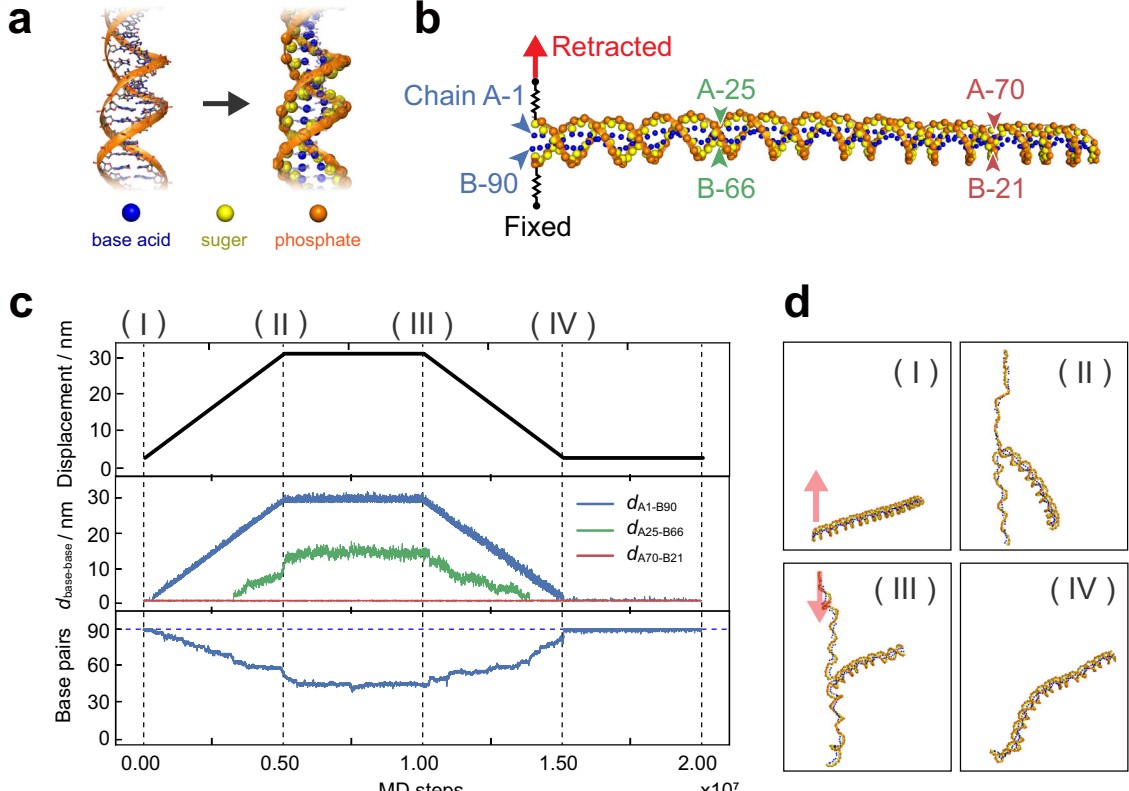

**Fig. 4 Course-grained molecular dynamics (MD) simulations. a** Schematic illustration of DNA modelling. Nucleotides are represented by three spherical sites corresponding to the nitrogenous base (blue), deoxyribose sugar (yellow), and phosphate group (orange). **b** Model of single-molecule junction of 90-mer DNA. Arrowheads indicate the selected bases. **c** Time course of displacement of moving end (top), the distance between complementary bases (middle), and the number of base pairs in the duplex region (bottom). **d** Snapshots of simulated structures of the zipper DNA. Labels (I)–(IV) correspond to MD steps shown in **c**.

**Self-restoring capability**. Highly feasible and reproducible formation of a single-molecule junction, in addition to high conductivity, is a critical step toward the realization of electronic devices using a single DNA molecule. The zipping configuration of the present molecular junction enabled us to employ a longer DNA duplex, which is known to improve the thermodynamic stability of the duplex in solution. Thus, we expect that the present strategy could improve the stability and/or the reproducibility of single-molecule junctions. To test this hypothesis, we investigated repeated formation of the single-molecule junction of the DNA zipper (Fig. 3a). An ssDNA molecule and its complementary strand were tethered to the STM tip and the Au(111) surface, similar to that in the *I*–*V* measurement. The molecular tip was first carefully brought in close proximity to the sample surface under the STM feedback loop. After awaiting for 0.3 s to facilitate hybridization and formation of the zipper structure, the STM tip was pulled up by 30 nm to record the conductance trace. The whole process was repeated to record consecutive *G*–*z* traces. For each trace, the dwell length was determined as the trace length between 1.4 and 2.4 m$G_0$ and between 0.14 and 0.16 m$G_0$ for the 90-mer and 10-mer zipper, respectively. These ranges were determined on the basis of the conductance values and their standard deviations from the STM-BJ measurements (see Fig. 1b). Figure 3b shows the time course of the dwell length obtained by successive 700 *G*–*z* traces for 10-mer and 90-mer DNA. The dwell length was then compared to the plateau length of the single-molecule junctions (dashed lines in Fig. 3b) of the 90-mer or 10-mer zipper DNA (Supplementary Note 1) to determine whether the molecular junction was successfully formed. It is clear that the 90-mer DNA zipper structure significantly enhances the repeatability of formation of the single-molecule junction compared with that of the 10-mer DNA zipper structure. The maximum number of repeated junction formations of the 90-mer DNA zipper reached 78; the single-molecule junction of the 90-mer DNA zipper was repeatedly reproduced for approximately 100 s despite the repeated perturbation of the junction by the tip displacements (Fig. 3c). The time course of the dwell length was quantitatively analyzed using joint probabilities (Supplementary Note 7). The analysis led to the same conclusion: the repeated and random formation of the molecular junction for 90-mer and 10-mer DNAs, respectively. The repeated formation of the DNA zipper junction was also confirmed by measuring mechanical forces exerted on the junction with AFM (Supplementary Note 8). A plausible model of DNA dynamics in this experiment was proposed in Fig. 3d. Repeated formation of the molecular junction for the 90-mer DNA could be due to partial preservation of the DNA duplex during pull-up procedures in the current measurements. This is not the case for the single-molecule junction of the 10-mer DNA zipper, since the pull-up distance of 30 nm is enough to break this junction considering the length of the duplex. The displacement dependence of the restoration behavior demonstrates the participation of the partially hybridized duplex and thus corroborates this model (Supplementary Note 9). The self-restoring behavior found for the present zipper junction opens up a way for reliable operations of single-molecule devices.

**MD simulations**. Course-grained MD simulations were performed to confirm partial preservation of the duplex for the single-molecule junction in the DNA zipper configuration. The

CafeMol software[42] with 3SPN.2 C model[43,44] was used for these calculations. It has been shown that this model successfully reproduces melting phenomena, persistence lengths, and major and minor grooves of dsDNA[43–45]. In the present work, one end of the 90-mer DNA duplex in the single-molecule junction was linked to two springs to represent the STM experiments (Fig. 4a, b). Then, the terminal base in one strand of the duplex (Chain A in Fig. 4b) was gradually pulled up by 30 nm in the simulation, as in the experiments (Fig. 3d). This base was then moved down to the initial position after the waiting time, as shown in the top panel of Fig. 4c. The distances between the selected complementary bases and the number of base pairs contained in the duplex region were plotted in the middle and bottom panels of Fig. 4c, respectively. Figure 4d shows selected snapshots of the single-molecule junction. Crucially, the DNA duplex is partially preserved after retraction (Fig. 4c, (II)), and the partial duplex, containing 44 base pairs, persisted after the waiting time (Fig. 4c, (III)). As the tethered end was pushed back to its original position after retraction, the unwound portion of DNA gradually reannealed to restore the complete duplex (Fig. 4c, (IV); see also Supplementary Movie 1). The distances between the complementary bases remained unchanged at the DNA end opposite to the tethered base (Fig. 4c, middle), demonstrating again the partial preservation of the DNA duplex. We carried out additional simulations, where the tethered end was released after the pull-up of 30 nm. The reannealing also occurred in this case, as found for the simulations above, and the complete duplex emerged (Supplementary Movie 2). These findings prove that spontaneous base-pairing accounts for duplex restoration, and not the external force exerted by the spring, which represents the STM tip in the experiments.

The lifetime of conventional single-molecule junctions, such as those of alkanedithiol, is typically short (in the order of milliseconds at room temperature), which poses fundamental problems for the realization of single-molecule electronics. These junctions break at the interface between the constituent molecule and the electrode, because the bonding between the functional group of the molecule and the electrode, e.g., Au–S bond, is weakest in the junction[46]. It has been reported that the force required to break the Au–S or Au–Au bond is 1–2 nN[47], while the DNA unzipping takes place under the force of 10–50 pN[22–24]. Thus, in the present junctions, the breakdown is attributed to be via DNA unzipping. The unzipping relaxes the tensile stress on the junction. Therefore, the duplex of long DNA zippers can be partially preserved, leading to prolonged preservation of the single-molecule connection with a DNA zipper. These results demonstrate that the DNA zipper offers a means to achieve both reproducible formation by the self-restoring capability and high conductivity.

In summary, we developed a DNA single-molecule junction where the duplex orthogonally clamped the metal nanogap. The electrical measurements based on STM revealed that the molecular junction of the DNA zipper attains high conductance, originating from the delocalized π system of the DNA near the electrodes. Moreover, we found an attractive self-restoring capability that the single-molecule junction can be repeatedly formed without breakdown of the whole structure even after an electrical failure. This advantage arises from the partial unzip of the DNA within the junction, which is supported by the coarse-grained MD simulations. It is also worth noting that conductance of the molecular junction exhibited no steep dependence on DNA length, which is distinctly different from the conventional single-molecule junction of DNA. Therefore, we expect that further functionalization could be possible through the use of a long DNA sequence as a scaffold to form conjugates with a wide range of functional (bio)molecules.

## Methods

**Sample preparation**. An Au(111) surface was prepared by thermal evaporation on a mica surface. Au wires (99.999%, 0.25 mm diameter) were chemically etched with 3.0 M NaCl solution in 1% perchloric acid (HClO₄) to prepare the STM tips. We used 90-mer DNA (Supplementary Table 1) modified with 1,3-propanethiol [–(CH₂)₃SH] linkers at the hydroxy group of the 3′ end as the probe. The complementary strand was modified with –(CH₂)₃SH linkers at the phosphate group of the 5′ end. As a control, we employed 10-mer DNA (Supplementary Table 1), which is partially the same as the 90-mer DNA, and –(CH₂)₃SH linkers were introduced at the 3′ end. The complement strand was modified with –(CH₂)₃SH linkers at the 5′ end. All the DNAs, synthesized by solid-phase synthesis using the phosphoramidite method[48], purified by high-performance liquid chromatography, and characterized by time-of-flight mass spectrometry (Supplementary Note 10), were purchased from Tsukuba Oligo Service (Ibaraki, Japan). For the preparation of the duplex, 1 μM solutions of ssDNAs were mixed with 10 mM phosphate-buffered saline (PBS) solution. This mixture was heated at 75 °C for 45 min. and slowly cooled to room temperature. The Au substrate was immersed in the solution for at least 2 h. For the experiments involving the molecular tip, STM tips were immersed in 1 μM ssDNA in PBS solution.

**STM measurement**. STM measurements were performed on an SPM 5100 system (Agilent Technologies, Santa Clara, CA, USA). The tunnelling current was sampled at 20,000 Hz. The bias voltage was 20 mV in all experiments, except for the *I–V* measurement. In the analyses, conductance traces with a simple exponential decay were removed based on an automated algorithm, in which the presence or absence of a plateau was judged using the conductance histogram constructed for each conductance trace[49]. Current measurements were repeated using independently prepared tips and sample surfaces. The reproducibility was confirmed by comparing conductance histograms obtained using every independent sample surface.

**Theoretical calculation**. We conducted the course-grained MD simulation with CafeMol[42] software. The 3SPN.2C model developed by the de Pablo group was used for DNA[44]. All the simulations were performed at 300 K. The DNA sequence as employed in the experiments was used for the simulation. For simulating the experiment in ambient conditions, no electrostatic interactions were taken into consideration. One terminus in each DNA strand was connected with springs for modelling the bridge between thiol linkers of DNA and Au atoms. The stiffness of the spring was 8.5 N/m, which is equal to the strength of the Au–Au bond[47,50].

**Reporting Summary**. Further information on research design is available in the Nature Research Reporting Summary linked to this article.

## Data availability

The data that support the findings of this study are available from the corresponding author upon reasonable request. Source data for Figs. 1–4 and Supplementary Fig. 1–12 are archived at https://doi.org/10.5281/zenodo.5515109. Source data are provided with this paper.

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

## Acknowledgements

This work was financially supported by Grants-in-Aid for Scientific Research (No. JP16K14018 and JP18H02003), JST CREST (No. JP-MJCR18I4), JSPS Fellows (No. 19J20605) from the Ministry of Education, Culture, Sports, Science and Technology (MEXT) of Japan, Iketani Science and Technology Foundation, and Murata Science Foundation.

## Author contributions

T.H., S.K., S.F., M.K., and T.N. conceived and designed the experiments. T.H., S.F., and Y.J. performed single-molecule measurements. T.H., T.T., and N.K. contributed to theoretical calculations. All the authors contributed to the analysis and interpretation of results and wrote the paper.

## Competing interests

The authors declare no competing interests.
