## [Peer Review File · Nature Communications]

Single-molecule junction spontaneously restored by DNA zipperEditorial Note: Parts of this Peer Review File have been redacted as indicated to maintain confidentiality.

Reviewers' comments:

Reviewer #1 (Remarks to the Author):

I recently reviewed this paper for another journal. As far as I can tell, the paper has not been modified in response to that earlier review. I would be happy see the author's response to my concerns. Here is that earlier review again:

The authors present a reasonably plausible case for the formation of DNA junctions attached to an STM probe by e.g., a 5' terminal and a substrate by a 3' terminal. The results are potentially interesting and deserving of publication. I note that the work has great similarities to the STM study of G tetraplexes by Sha et al (NNT, 13, 316). This paper is not cited. It must be used as context for the present work.

The main flaw of the paper as it stands is the lack of obvious corroborating experiments. The authors review AFM measurements on the formation of hybridized structures, but make a weak case based on STM break junction data alone. I could not find a discussion comparing the conductances measured with IV curves (Figure 2) to the conductances measured with break junctions (Figure 1), surely the most basic piece of potentially supporting information?

The preferred method would be conducting AFM (c-AFM) measurements in which conductance values were obtained at the same time as the (already well characterized) single molecule unzipping force curves are measured.

If the authors have no access to c-AFM, then at least they could confirm the hybridization kinetics inferred from the STM measurements using force curve measurements on an AFM.

Finally, another approach for obtaining simultaneous force and conductance measurements can be found in the elegant work of Tao who used a small modulation of the Z PZT with lockin detection to measure the junction stiffness over a pulling experiment.

Reviewer #2 (Remarks to the Author):

The manuscript "Single-molecule junction spontaneously restored by DNA zipper," by Harashima et al. describes single-molecule junction experiments on DNA where the DNA is placed in the transverse direction to transport (instead of the longitudinal direction) as is typically done in break-junction experiments. They study two cases, a 10-mer DNA duplex and a 90-mer DNA duplex in this configuration, and conclude that if the molecule is long enough then it does not fully unzip during pulling experiments, and can re-hybridize during experiments. This conclusion is potentially interesting, however, I feel the manuscript does not sufficiently prove the point at this juncture, and as such would not be in favor of publication without significantly more experimental work.

First, the point that hybridization is occurring in situ and remaining partially hybridized during stretching should be better supported. A series of experiments with different DNA lengths and different pulling distances would support this argument.

Second, the I-V's and TVS data don't fully support the argument. Why was TVS not done on pre-hybridized samples as well? Do the slopes of the IVs correspond to the conductance values found in Fig. 1? It is impossible to say that the value obtained is from the duplex in this case.

The discussion of why the 90-mer is more conductive than the 10mer needs to be expounded on. Are there difference in the synthesis process for these different lengths? Some processes result in different end chemistries. The 90-mer is ~10x higher in conductance, which is surprising given the transport distances should be the same (tunneling across the end)? Or do the authors envision some other process? Additional lengths/sequences in between these two extremes may help with this conclusion. At the minimum, to support the current claim, they would need to provide some evidence that the delocalization they mention is greater than 10 basepairs, which would be surprising.

The linker chemistry is not described in detail. Are the linkers attached to the phosphate groups? Often the synthetic routes for the 3' and 5' are different and there are differences in the length of the saturated chain in between. How are they able to get around this issue? Are these synthesized in house? Or purchased?

The statistical analysis is not given in sufficient detail. Are all traces included in Fig. 1? It is also common to plot these in log scale (or semi-log).

Reviewer #3 (Remarks to the Author):

In this manuscript, the authors report measurements of single-molecule DNA junctions, showing that a "zipper" configuration can be used to enhance stability of the molecular junction. This work is innovative and of interest to a broad range, yet some critical comments prevent me from giving a positive recommendation. These must be addressed by the authors before I can give a final recommendation.

1. The authors claim to (and describe the protocol of) preparing DNA wires which are treated edge linkers at the same direction ('3 and '5 in the complementary strands). The authors must bring conclusive evidence that the linkers are indeed where they are supposed to be (the minimum would be a reference to show that this process works, but even this was not supplied by the authors).
2. The authors use random DNA sequences. Why? Randomness makes the system much harder to understand, and also makes the comparison between the 10-mer and 90-mer junctions essentially irrelevant (because disorder and the sequence dramatically affect the electronic structure of the junctions).
3. The authors discuss the β -values of the junction, comparing them to reported β values for direct tunneling from the literature. Why isn't there a measurement in this setup? β values can differ between experiments, electrodes, etc.
4. The authors relate between the β values and the electronic gap (more precisely the gap between the Fermi level and the frontier orbital energy). It is unclear why the orbital energy changes with distance if transport (and hence the distance stretch) is perpendicular to the strands. The authors should provide at least a simple (LCAO) model for this change, and explain the observations. Otherwise, one cannot learn anything about the junction from this measurement.
5. Also - because the DNA is disordered, this also affects the gap, again implying that a comparison with other systems is meaningless (or at least should be corroborated by some calculation).
6. The TVS data is very noisy. Specifically, very few measurements were taken, and the peak at 0.65eV seems to be almost as large as the peak at 0.4-0.5 eV. Why so few measurements? where is the discussion on this second peak?
7. Why is it reasonable to compare these data with the data of an 8-mer DNA junction, if there is disorder?
8. The main result is that, basically, stretching does not fully "un-zipp" the DNA. This makes comment #1 even more important.
9. Why did the authors stop at 10 and 90 base pairs? a much more useful study would be to see the junction restoration effect for junctions with 10,20,30,...90 base pairs. What is the minimal length of DNA that can be used to stabilize the junction this way?

In short, while this project is very interesting and exciting, the data presented in this paper is in my opinion too preliminary, and the paper cannot be accepted in its present form. The authors are welcomed to add to this paper and respond to my comments.

List of Revisions

Single-molecule junction spontaneously restored by DNA zipper

Manuscript ID: NCOMMS-19-41904

Takanori Harashima, Shintaro Fujii, Yuki Jono, Tsuyoshi Terakawa, Noriyuki Kurita, Satoshi Kaneko, Manabu Kiguchi, and Tomoaki Nishino

First, we sincerely appreciate the time and effort each reviewer has spent providing insightful feedback to help us improve our manuscript. We have incorporated changes that reflect the detailed suggestions graciously provided. We hope that our revisions and the responses provided as follows satisfactorily address all the issues and concerns that have been noted.

To facilitate your review of our revisions, the following is a point-by-point response to the questions and comments.

Reviewer #1

I recently reviewed this paper for another journal. As far as I can tell, the paper has not been modified in response to that earlier review. I would be happy see the author's response to my concerns. Here is that earlier review again:

Comment 1. *The authors present a reasonably plausible case for the formation of DNA junctions attached to an STM probe by e.g., a 5' terminal and a substrate by a 3' terminal. The results are potentially interesting and deserving of publication. I note that the work has great similarities to the STM study of G tetraplexes by Sha et al (NNT, 13, 316). This paper is not cited. It must be used as context for the present work.*

Response. As per the comment, the recommended paper was cited as reference 13 and is described in the Introduction as follows.

(Main text, Page 3, Lines 7–12)

“Many studies have revealed that there is a change in the electron transport properties of DNA after ligand binding⁷⁻⁹. These include an intercalator whose perturbation of base stacking leads to switching or rectifying properties of DNA¹⁰⁻¹². Moreover, a three-terminal DNA connected by a guanine quadruplex has been exploited to construct a single-molecule junction, and it was demonstrated that the intricate junction structure shows excellent functionality for a charge splitter¹³.”

Comment 2. *The main flaw of the paper as it stands is the lack of obvious corroborating experiments.*

The authors review AFM measurements on the formation of hybridized structures, but make a weak case based on STM break junction data alone. I could not find a discussion comparing the conductances measured with IV curves (Figure 2) to the conductances measured with break junctions (Figure 1), surely the most basic piece of potentially supporting information?

Response. Following this comment, we compared the conductance values measured with the I - V curves and the BJ technique and found that they are consistent with each other. A detailed discussion was added to the Supplementary Information, and a concise description was included in the main text as follows.

(Main text, Page 7, Lines 12–15)

“The high-conductance state was thus ascribed to the molecular junction of the DNA zipper. The conductance as estimated by the I - V properties agrees with the conductance as determined by the BJ studies (Fig. 1c and d, see Supplementary Note 4), supporting the assignment of a high-conductance state to the DNA junction.”

(Supplementary information, Supplementary Note 4, Pages 7 and 8)

“Supplementary Note 4: Conductance of molecular junction as estimated by I - V curves.

To confirm the formation of the 90-mer DNA zipper junction in the I - V measurements as presented in Fig. 2a–c, we compared the conductance derived from the I - V properties and the conductance of the junction found in the STM-BJ experiments (Fig. 1). First, the I - V curves were clustered into two distributions using the fuzzy c -means algorithm^{5,6}, since two separate distributions were found in the V_{trans} histogram (Fig. 2). The successful classification is evident from the distinct difference between the I - V properties of the resultant sub-clusters (Supplementary Fig. 4a and b). The transition voltage analysis indicated that the high-conductance state corresponds to the DNA zipper junction, and the low-conductance state is ascribed to the tip-substrate gap without the molecular bridge. The conductance of each I - V curve in the sub-clusters was calculated, and the statistically most probable conductance values, G_{IV} , were found to be 3.1 and 4.0 mG_0 for the low- and high-conductance states, respectively, based on the histogram analysis (Supplementary Fig. 4c). In comparing these conductance values with corresponding values obtained by the STM-BJ study, care must be taken due to differences in the experimental conditions. Specifically, the tip-substrate distance was much smaller for the I - V measurements than for the STM-BJ measurements. This is because the present protocol to acquire the I - V curves is based on the so-called I - t measurements⁷, in which the STM tip is brought in close proximity to the sample surface to facilitate the spontaneous formation of the molecular junction⁸. Direct tunnelling between the metal tip and substrate is non-negligible under this condition, as evidenced by the relatively large conductance for the low-conductance state. We thus eliminated the contribution of direct tunnelling by considering the difference between the

conductance of the high- and low-conducting states. The conductance of the molecular junction is consequently estimated to be 0.9 mG₀. This value is consistent with the conductance as determined by the STM-BJ experiments, which supports the assignment of the high-conductance state of the I - V property that arises due to the formation of the DNA zipper junction.”

Comment 3. *The preferred method would be conducting AFM (c-AFM) measurements in which conductance values were obtained at the same time as the (already well characterized) single molecule unzipping force curves are measured. If the authors have no access to c-AFM, then at least they could confirm the hybridization kinetics inferred from the STM measurements using force curve measurements on an AFM. Finally, another approach for obtaining simultaneous force and conductance measurements can be found in the elegant work of Tao who used a small modulation of the Z PZT with lockin detection to measure the junction stiffness over a pulling experiment.*

Response. Following this comment, we performed additional AFM experiments. We found that the c-AFM measurements were difficult to obtain in the present case, despite lengthy and dedicated work under collaboration with a specialist [redacted]. For precise simultaneous measurements of the force and conductance, an aqueous environment was necessary. However, under this environment, a large background current, stemming from electrochemical processes, prevented the conductance measurements. We thus used the force curve measurement, suggested as the second option by the Reviewer, to corroborate the self-restoring property of the DNA zipper junction. The results are described in the Supplementary Information, and they are summarized in the main text as follows.

(Main text, Page 10, Lines 20 and 21)

“The repeated formation of the DNA zipper junction was also confirmed by measuring mechanical forces exerted on the junction with AFM (Supplementary Note 7).”

(Supplementary information, Supplementary Note 7, Pages 14–16)

“Supplementary Note 7: AFM measurements of repeated formation of DNA zipper junction.

The self-restoration behavior of the DNA zipper junction was characterized by atomic force microscopy (AFM). An Au(111) surface modified with the double-stranded 90-mer DNA served as the sample surface. We first investigated the force–distance curves with an Au-coated cantilever (Supplementary Fig. 8a). The distinct rupture force demonstrated the successful formation of the DNA zipper junction between the tip and substrate, as discussed in the main text with a series of STM-based studies. A close inspection of the force–distance curves revealed the stepwise decrease in force by approximately 50 pN immediately before the complete rupture of the molecular junction (Supplementary Fig. 8b). Sawtooth-like force changes were also observed during the extension before the complete rupture

(Supplementary Fig. 8c). We attributed these discrete changes to the force-induced melting of DNA duplexes in the junction. In fact, the statistically most probable force of 0.03 N found by the histogram analysis (Supplementary Fig. 8d) is consistent with the reported value for DNA melting¹⁸⁻²⁰. The force curves measured in the present study are thus interpreted as a series of DNA melting from the molecular junction that initially contained hundreds of DNA duplexes.

Next, we addressed the repeatability of the formation of the zipper DNA: the force–distance curves were measured with an extension of 30 nm, which is insufficient to fully unzip the DNA duplex. The contour length of the 90-mer DNA is estimated to be 29 nm by assuming a rise distance of 0.33 nm/base-pair, but extensions longer than this contour length are necessary to mechanically melt the DNA duplex under the present linking configuration (see Fig. 1a in the main text). No complete rupture was observed in the resulting force–distance curves as expected (Supplementary Fig. 9a). Importantly, repeated approach–retract cycles resulted in force curves very similar to each other, indicating the restoration of the zipper DNA duplex (Supplementary Fig. 9b). The restoration behavior was quantitatively assessed by its probability defined by $1-(F_{\text{re}}-F_{\text{app}})/F_{\text{re}}$, where F_{re} and F_{app} are the force values at the displacement of 15 nm in the retraction and the subsequent approach processes, respectively. The histogram shows that 82% of the DNA zipper was restored during the re-approach after an extension of 30 nm (Supplementary Fig. 9c). A similar quantity was derived from the STM-BJ study. The restoration probability was directly measured by the occurrence of the successive formation of the molecular junction in the consecutive G – z measurements as presented in Fig. 3b and 3c in the main text. The probability was 77%, which is consistent with that found in the AFM study as argued above. In contrast, the restoring behavior was absent when an extension of 60 nm was applied in the approach–retract cycles. The results corroborate the repeated formation of the DNA zipper junction by self-restoration as observed in the STM-BJ experiments.”

Reviewer #2

The manuscript “Single-molecule junction spontaneously restored by DNA zipper,” by Harashima et al. describes single-molecule junction experiments on DNA where the DNA is placed in the transverse direction to transport (instead of the longitudinal direction) as is typically done in break-junction experiments. They study two cases, a 10-mer DNA duplex and a 90-mer DNA duplex in this configuration, and conclude that if the molecule is long enough then it does not fully unzip during pulling experiments, and can re-hybridize during experiments. This conclusion is potentially interesting, however, I feel the manuscript does not sufficiently prove the point at this juncture, and as such would not be in favor of publication without significantly more experimental work.

Comment 1. *First, the point that hybridization is occurring in situ and remaining partially hybridized*

during stretching should be better supported. A series of experiments with different DNA lengths and different pulling distances would support this argument.

Response. As per this comment, we performed the requested experiments. The results are concisely described in the main text and discussed in detail in the Supplementary Information as follows.

(Main text, from Page 10, Line 22 to Page 11, Line 3)

“Repeated formation of the molecular junction for the 90-mer DNA could be due to partial preservation of the DNA duplex during pull-up procedures in the current measurements. This is not the case for the single-molecule junction of the 10-mer DNA zipper, since the pull-up distance of 30 nm is enough to break this junction considering the length of the duplex. The displacement dependence of the restoration behavior demonstrates the participation of the partially hybridized duplex and thus corroborates this model (Supplementary Note 8).”

(Supplementary information, Supplementary Note 8, Pages 17–19)

“Supplementary Note 8: Displacement dependence of restoration capability of DNA zipper junction.

We investigated the relationship between the restoration capability of the DNA zipper junction and the tip displacement. The G - z measurements were performed with the STM tip modified with single strands of the DNA zipper. An Au(111) substrate covered with the complementary single strands served as the sample. The DNA tip was brought very close to the sample surface and pulled up to record the G - z traces. The displacements during the tip retraction were consecutively varied (10, 20, 40, 80, and 120 nm for the 90-mer DNA zipper; or 5, 10, 20, 40, and 80 nm for the 10 mer), and consequently, a single dataset contains five traces having different displacements. A representative example of the sequence of the G - z traces measured with the 90-mer DNA zipper is shown in Supplementary Fig. 10a. The presence or absence of the molecular junction of the DNA zipper was assessed based on the dwell length, i.e., the trace length between 1.4 mG₀ and 2.4 mG₀ and between 0.14 mG₀ and 0.16 mG₀ for the 90-mer and 10-mer zipper, respectively (see Supplementary Fig. 10b). The restoration capability was then quantitatively evaluated using a correlation coefficient of the dwell lengths taken from the two consecutive G - z traces. For example, the correlation coefficient for the 20-nm displacement, C_{20} , was calculated as

$$C_{20} = \frac{\sum_n \left(\Delta L_{20}(n) \Delta L_{40}(n) \right)}{\sqrt{\sum_n \left(\Delta L_{20}(n) \right)^2 \sum_n \left(\Delta L_{40}(n) \right)^2}}$$

, where

$$\Delta L_x(n) = L_x(n) - \frac{\sum_n L_x(n)}{N}$$

, and n , N , and $L_x(n)$ denote the index of the dataset, total number of the dataset, and the dwell length for the x nm displacement in the n th dataset, respectively. In the above example, the correlation coefficient for the 20-nm displacement involved the dwell length for the 40-nm displacement, because the restoration after the 20-nm displacement was reflected in the next $G-z$ trace, i.e., the trace for the 40-nm displacement. The correlation coefficients for the other displacements were similarly defined and calculated. In the case of the displacement of 120 nm for the 90-mer DNA zipper (or 80 nm for 10-mer), whose traces were located at the end of each dataset, the dwell length of the first $G-z$ trace (10-nm displacement for 90-mer, or 5-nm displacement for 10-mer) in the next dataset was used, along with the dwell length of the 120-nm (or 80-nm) displacement, for the calculation of the correlation coefficient. Supplementary Fig. 10c shows the resulting correlation coefficients of the 10-mer and 90-mer DNA zippers. For the 90-mer zipper, we found high correlation coefficients for shorter displacements and a gradual decrease for longer displacements. In contrast, in the case of the 10-mer zipper, the correlation coefficient was high only for the shortest displacement and decreased rapidly. The high correlation at the shorter displacements means that the DNA zipper junctions were regenerated during the two consecutive $G-z$ measurements, demonstrating the restoration capability. It is noteworthy that the restoration behavior was observed even for the junction with 10-mer DNA, though at the short displacement. We thus anticipate that dsDNAs stable at room temperature can induce the restoration effect based on the present strategy.

The mechanism of the junction restoration can be inferred based on the observed displacement dependence. The maximum displacements for the DNA zipper junction without completely breaking the double strand can be estimated to be approximately twice the length of the DNA under the present junction structure (Supplementary Fig. 10d), i.e., 5.9 and 59 nm for the 10-mer and 90-mer DNA zippers, respectively. It is apparent from Supplementary Fig. 10c that the correlation was lost, and the coefficients decreased to the common background value of approximately 0.3 when the displacement exceeded the maximum tolerable distances. The gradual decrease in the correlation for the 90-mer DNA zipper junction until complete double-strand dissociation clearly indicates participation of the partially hybridized structures of the DNA zipper in the restoration behavior. In the $G-z$ measurements with moderate displacements, the DNA zipper junction remained partially hybridized during the tip retraction. This structure facilitates the regeneration of the complete junction by re-hybridization²⁴⁻²⁶. The non-zero background of the correlation coefficient after the breakdown of the zipper junction is most probably attributed to the use of the dwell length for their calculation, because the dwell length remained at a finite value even in the absence of the junction, which was due to direct electron tunnelling between the tip and substrate.”

Comment 2. Second, the $I-V$'s and TVS data don't fully support the argument. Why was TVS not done on pre-hybridized samples as well? Do the slopes of the IVs correspond to the conductance values found in Fig. 1? It is impossible to say that the value obtained is from the duplex in this case.

Figure R1. Formation of DNA zipper junction for pre-hybridized dsDNA on substrate (top) and for molecular tip with ssDNA-modified substrate (bottom).

Response. Regarding the first question, we did not present the TVS data for the pre-hybridized sample because of the low probability of junction formation in the $I-V$ measurements. The present single-molecule junction shows a unique dynamic behavior, i.e., self-restoration, and thus a constant tip-sample distance should be ensured for every $I-V$ curve to infer its intrinsic electronic properties. The widely employed $I-V$ measurements based on the STM-BJ protocol cannot meet this requirement, and we employed a protocol where the STM tip was held stationary at the predetermined distance from the sample surface to measure the molecular junctions that emerged by spontaneous bridging of the tip and substrate. We found that the junction was hardly formed for the pre-hybridized DNA, most probably due to the close proximity of the two anchoring groups at the end of the duplex (Figure R1). Therefore, a molecular tip approach, where the junction was formed by *in situ* hybridization between the tip and substrate, was utilized to acquire the TVS.

As for the second question, the $I-V$ slopes were consistent with the conductance of the duplex as seen in Fig. 1, thereby demonstrating that we successfully detected the duplex in the $I-V$ study. This comment is the same as Comment 2 from Reviewer #1. Please see its Response for the detailed revisions.

Comment 3. The discussion of why the 90-mer is more conductive than the 10mer needs to be expounded on. Are there difference in the synthesis process for these different lengths? Some processes result in different end chemistries. The 90-mer is $\sim 10x$ higher in conductance, which is surprising given the transport distances should be the same (tunneling across the end)? Or do the authors envision some other process? Additional lengths/sequences in between these two extremes may help with this conclusion. At the minimum, to support the current claim, they would need to provide some evidence that the delocalization they mention is greater than 10 basepairs, which would be surprising.

Response. Regarding the first question about the synthetic process, all the DNAs were synthesized by the same protocol. Thus, the end chemistry is common for DNAs having different lengths (please also see the response to the next question).

As for the second and third questions about the conductance, the difference between the 10- and 90-mer DNAs is surprising, but we do not envision other processes than tunnelling across the end. Following this comment, we newly measured the conductance of the DNA zipper junctions of another sequence (*revision i*) and additional lengths (30-, 50-, and 70-mers, in addition to the 10- and 90-mers originally reported in the manuscript; *revision ii*), and consequently confirmed our conclusion that the conductance increased with the increase in the DNA length.

Finally, concerning delocalization, we performed *ab initio* molecular orbital calculations for DNAs of different lengths (*revision iii*). We found a decreased HOMO-LUMO gap for longer DNA, highlighting the effect of the delocalized π -electrons over the DNA duplex. The results and a discussion of the newly added experiments and calculations were added to the main text and the Supporting Information as follows.

Revision i) conductance of the DNA zipper junction with another sequence

(*Main text, Page 5, Lines 15–17*)

“Further STM-BJ experiments showed that the conductance of the molecular junction reflects the DNA sequence (Supplementary Note 2), indicating that the electron transport is mediated by the DNA.”

(*Supplementary information, Supplementary Note 2, Page 4*)

“**Supplementary Note 2: Effect of base sequence on conductance of DNA zipper junction.**”

We investigated the effect of the DNA sequence on the conductance of the single-molecule junction with the zipper configuration. A portion of the sequence of the sample 90-mer DNA was changed: the 30 bases located at the thiolated terminal were replaced with the base sequence of (A₅T₅)₃. STM-BJ measurements were carried out using an Au(111) substrate modified with this “mutated” DNA. The *G*-*z* traces exhibited plateaus (Supplementary Fig. 2a), demonstrating the formation of the DNA zipper junction. The statistically most probable conductance of the junction was found to be 0.13 mG₀, based on the histogram analysis of the traces (Supplementary Fig. 2b). Notably, this value was smaller than the conductance for the junction of the original 90-mer DNA (1.9 mG₀). The smaller conductivity of AT-rich DNAs as compared with that of GC-rich counterparts is commonly observed^{6,7}. The prominent effect of the base sequence on the junction conductance proves that the electron transport as measured in the present study involves the orbitals of the DNA perpendicularly trapped in the electrode gap.”

Revision ii) conductance of the DNA zipper junction with additional lengths

(Main text, Page 6, Lines 1–3)

“We measured the conductance of the zipper junction of a variety of DNA lengths ranging from 10 to 90 base pairs and confirmed that the conductance value increased as the DNA length increased (Supplementary Note 3).”

(Supplementary information, Supplementary Note 3, Pages 5 and 6)

“Supplementary Note 3: DNA length dependence of conductance of zipper junction.

We measured the conductance of zipper junctions composed of 30-, 50-, and 70-mer DNA, in addition to the 10- and 90-mer DNA reported in the main text, by the STM-BJ technique to investigate the effect of DNA length on electron transport properties. For all of the investigated DNAs, a single peak appeared in the conductance histograms (Supplementary Fig. 3a), indicating the successful formation of zipper junctions (see Fig. 1a in the main text). Importantly, the conductance value increased as the DNA length increased (Supplementary Fig. 3b). The observed length dependence is in stark contrast with that of conventional single-molecule junctions of DNA. Most of the DNA junctions bear two linker groups, such as –SH, at the opposite terminals of the DNA. Electron transport steeply attenuates with increased DNA length in these cases, since the electrons travel through the whole duplex. However, in the present work, the linker groups were introduced at the same end of the DNA, and no such attenuation of electron transport happened. The increased conductance can be explained based on transition voltage spectroscopy (Fig. 2d in the main text) and theoretical calculations (Supplementary Note 5).”

Revision iii) *ab initio* molecular orbital calculations for DNAs of different lengths

(Main text, Page 8, Lines 8–16)

“We attribute the small V_{trans} , that is, the decreased energy gap, to delocalization of the π -orbitals of DNA over its long base pairs. To prove this, molecular orbital calculations were performed based on density functional theory (Supplementary Note 5). We indeed found that the energy gap between the highest occupied molecular orbital (HOMO) and the lowest unoccupied molecular orbital (LUMO) decreased with the increase in DNA length, in line with previous theoretical studies^{36,37}. The larger conductance of the 90-mer DNA junction compared to the 10-mer counterpart (Fig. 1c) is consistent with the length-dependent decrease in the HOMO-LUMO gap. Thus, we conclude that the single-molecule junction of the DNA zipper attains high conductance due to the delocalized π system of the stacked DNA bases near the electrodes.”

(Supplementary information, Supplementary Note 5, Pages 9–11)

“Supplementary Note 5: Theoretical calculation of electronic structure of DNA.

The electronic structure of double-stranded DNA (dsDNA) was theoretically investigated by means of density functional theory (DFT). The initial structures for the DFT calculations were prepared by molecular dynamics (MD) simulations using the AMBER package (version 16)⁹. We prepared 10-, 15-, 20-, 25-, and 30-mer DNAs with the same sequences as in our conductance experiments. The BSC1 force field¹⁰ was used to describe DNA force field parameters, and Na⁺ ions were placed at a position equally spaced from the two oxygen atoms of the phosphate group of the DNA backbone¹¹. All MD simulations were performed with the implicit solvent based on the generalized Born model¹². After the initial minimization, the equilibration processes were executed for 10 ns. The structures with the minimum total energies were extracted for the DFT calculation (Supplementary Fig. 5). The DFT calculation was performed with a double-numeric polarized basis set as the electronic wave function with a real-space cutoff of 0.5 nm using Dmol³ code. The generalized gradient approximation (GGA) was used in the scheme of Perdew-Burke-Ernzerhof (PBE) to describe the exchange-correlation (XC) functional¹³. The solvation effects were considered based on the conductor-like Screening model (COSMO)^{14,15}, and a dielectric constant of 78.54 was used for solvation by water.

Supplementary Fig. 6 shows the energy-level diagram of the molecular orbitals (MOs) of dsDNA. It can be clearly seen that energy levels of the highest occupied MO (HOMO, E_{HOMO}) and the lowest unoccupied MO (LUMO, E_{LUMO}) increase and decrease, respectively, with the increase of the DNA length. The E_{HOMO} (or E_{LUMO}) of 30-mer DNA was increased by 0.079 eV (or decreased by 0.226 eV) compared to 10-mer DNA. Within the resonant level model, the electronic conductance becomes large, as the frontier orbital responsible for electron transport (either HOMO or LUMO in general) is energetically close to the Fermi level (E_{F}) of the electrodes¹⁶. The increase in the single-molecule conductance of the zipper junction with the longer DNA observed in the experiments (Fig. 1b in the main text and Supplementary Fig. 3b) is thus attributed to the increased E_{HOMO} and decreased E_{LUMO} , given that E_{F} lies in the HOMO–LUMO gaps. These results are consistent with previous reports in which DNA bases with higher HOMO levels exhibit higher electrical conductance¹⁷.”

Comment 4. *The linker chemistry is not described in detail. Are the linkers attached to the phosphate groups? Often the synthetic routes for the 3' and 5' are different and there are differences in the length of the saturated chain in between. How are they able to get around this issue? Are these synthesized in house? Or purchased?*

Response. The linkers were attached to the phosphate and hydroxy groups in the modification of the 5' and 3' terminals, respectively. The synthetic route was the same, i.e., *via* solid-phase synthesis using the phosphoramidite method in both cases, and common $-(\text{CH}_2)_3-$ chains were included between the phosphate (5' end) and hydroxy (3' end) groups and the thiol (Scheme R1). The modified DNAs were purchased. We have revised the manuscript to describe the linker chemistry in detail as follows.

(Main text, From Page 14, Line 24 to Page 15, Line 8)

“We used DNA (5'-GCG CAA TGA AAG CCC GTG CCG TTA TCA GGC CGG ATT AGG TTA GAA TCG TGG AGC CAT TCC ACA TCC GCT TGT GGT TTG ACG GCC ACC-3') modified with 1,3-propanethiol $[-(\text{CH}_2)_3\text{SH}]$ linkers at the **hydroxy group of the 3' end** as the probe. The complementary strand was modified with $-(\text{CH}_2)_3\text{SH}$ linkers at the **phosphate group of the 5' end**. As a control, we employed 10-mer DNA. The sequence was 5'-GAC GGC CAC C-3', which is partially the same as the 90-mer DNA, and $-(\text{CH}_2)_3\text{SH}$ linkers were introduced at the 3' end. The complement strand was modified with $-(\text{CH}_2)_3\text{SH}$ linkers at the 5' end. **All the DNAs, synthesized by solid-phase synthesis using the phosphoramidite method⁴⁴, purified by high performance liquid chromatography, and characterized by time-of-flight mass spectrometry (Supplementary Note 9), were purchased from Tsukuba Oligo Service (Ibaraki, Japan).”**

Thiol modification of 5' terminal

Thiol modification of 3' terminal

Scheme R1. Solid-phase synthesis of DNA using the phosphoramidite method.

Comment 5. *The statistical analysis is not given in sufficient detail. Are all traces included in Fig. 1? It is also common to plot these in log scale (or semi-log).*

Response. Not all traces were included in Fig. 1. We revised the manuscript and added the relevant literature as reference 45 to show this procedure and the statistical analyses.

(Main text, Page 15, Lines 8–19)

“In the analyses, conductance traces with a simple exponential decay were removed based on an automated algorithm as reported in the literature⁴⁵. Current measurements were repeated using independently prepared tips and sample surfaces. The reproducibility was confirmed by comparing conductance histograms obtained using every independent sample surface.”

We further revised Fig. 1 to plot the histogram on a semi-logarithmic scale as per the above comment. In addition, representative traces were included for better presentation. The number of traces is now explicitly described in the caption of Fig. 1. These changes are quoted below.

(Main text, Page 15, Fig. 1)

(Main text, Page 15, caption of Fig. 1)

“**Fig. 1** Single-molecule junction of DNA zipper. **a** Schematic illustration of the scanning tunnelling microscopy–break-junction (STM-BJ) measurements. **b** Representative conductance traces for unmodified Au(111) substrate (gray), and 90-mer and 10-mer DNAs (blue and purple, respectively). **c and d** 2D histograms of the conductance–displacement (G - z) traces for 90-mer and 10-mer DNAs, respectively. 2219 and 2635 traces were analyzed for histograms in c and d, respectively. Tip velocity, 31 nm/s; bias voltage, 20 mV.”

Reviewer #3

In this manuscript, the authors report measurements of single-molecule DNA junctions, showing that a "zipper" configuration can be used to enhance stability of the molecular junction.

This work is innovative and of interest to a broad range, yet some critical comments prevent me from

giving a positive recommendation. These must be addressed by the authors before I can give a final recommendation.

Comment 1. *The authors claim to (and describe the protocol of) preparing DNA wires which are treated edge linkers at the same direction (3' and 5' in the complementary strands). The authors must bring conclusive evidence that the linkers are indeed where they are supposed to be (the minimum would be a reference to show that this process works, but even this was not supplied by the authors).*

Response. As per the comment, time-of-flight mass spectra and their discussion was added to the Supporting Information as Supplementary Note 9. The relevant reference was newly cited as reference 44 in the main text. The revised text is as follows.

(Main text, Page 15, Lines 6–8)

“All the DNAs, synthesized by solid-phase synthesis using the phosphoramidite method⁴⁴, purified by high performance liquid chromatography, and characterized by time-of-flight mass spectrometry (Supplementary Note 9), were purchased from Tsukuba Oligo Service (Ibaraki, Japan).”

(Supplementary information, Supplementary Note 9, Page 20)

“Supplementary Note 9: Time-of-flight mass spectrum of synthesized DNA.

The DNA duplex was tethered to the electrodes (the STM tip and substrate) at the same end to create the zipper junction in the present work (Fig. 1a in the main text). To do so, the thiol-containing linker, that is, 1,3-propanethiol [-(CH₂)₃SH], was introduced at the hydroxy group of the 3' end of one of the strand. The same linker was also introduced at the phosphate group of the 5' end of another strand. These DNAs, synthesized according to the solid-phase synthesis using the phosphoramidite method and purified by high-performance liquid chromatography, were purchased from Tsukuba Oligo Service (Ibaraki, Japan). The purified products were characterized by time-of-flight mass spectrometry (TOF MS). Supplementary Fig. 11 shows TOF-MS spectra for the 90-mer strands. The measured mass of the molecular ion was consistent with the expected one, showing the successful introduction of the linker.”

Comment 2. *The authors use random DNA sequences. Why? Randomness makes the system much harder to understand, and also makes the comparison between the 10-mer and 90-mer junctions essentially irrelevant (because disorder and the sequence dramatically affect the electronic structure of the junctions).*

Response. We used the term “random” to indicate that the DNA sequence is not a simple repetition of a particular base (such as poly(dG)·poly(dC)). The DNAs employed in the present study have definite sequences as described in the Methods. Moreover, the sequence of the 10-mer DNA is common to part of the 90-mer DNA, which facilitates the comparison between the junctions. As per the comment, we

replaced the phrase “a random sequence” with “a sequence” in the manuscript to make the content clearer. The resultant description is as follows.

(Main text, From Page 14, Line 24 to Page 15, Line 1)

“We used DNA (5'-GCG CAA TGA AAG CCC GTG CCG TTA TCA GGC CGG ATT AGG TTA GAA TCG TGG AGC CAT TCC ACA TCC GCT TGT GGT TTG ACG GCC ACC-3') modified with 1,3-propanethiol [-(CH₂)₃SH] linkers”

Comment 3. *The authors discuss the β -values of the junction, comparing them to reported β values for direct tunneling from the literature. Why isn't there a measurement in this setup? β values can differ between experiments, electrodes, etc.*

Response. Following this comment, we measured the β values for the direct tunnelling. The results are included in Supplementary Fig. 1 in Supplementary Note 1. The relevant description in the main text was revised accordingly as follows.

(Main text, Page 5, Lines 9–11)

“The β_2 value found here is consistent with that for **direct tunnelling between the tip and substrate without the molecular junction (2.2 Å⁻¹, Supplementary Note 1).**”

Comment 4. *The authors relate between the beta values and the electronic gap (more precisely the gap between the Fermi level and the frontier orbital energy). It is unclear why the orbital energy changes with distance if transport (and hence the distance stretch) is perpendicular to the strands. The authors should provide at least a simple (LCAO) model for this change, and explain the observations. Otherwise, one cannot learn anything about the junction from this measurement.*

Response. The same issue was pointed out in Comment 3 by Reviewer #2. Please see its response above.

Comment 5. *Also - because the DNA is disordered, this also affects the gap, again implying that a comparison with other systems is meaningless (or at least should be corroborated by some calculation).*

Response. As described in the Response to Comment 1 of Reviewer #3, the sample DNAs have a definite sequence, rendering the comparison valid.

Comment 6. *The TVS data is very noisy. Specifically, very few measurements were taken, and the peak at 0.65eV seems to be almost as large as the peak at 0.4-0.5 eV. Why so few measurements? where is the discussion on this second peak?*

Response. Following this comment, we repeated the measurements to make the statistics more robust, and the TVS data shown in Fig. 2 of the main text were replaced. The data now show a single peak at 0.4–0.5 eV. The captions were accordingly changed to reflect these changes, as quoted below.

(Main text, Page 9, Fig. 2)

(Main text, Page 9, caption of Fig. 2)

“**Fig. 2** Current–voltage (I – V) and transition voltage spectra. I – V 2D histograms and V_{trans} histograms obtained with (a, d) molecular tip and modified substrate, (b, e) molecular tip and bare substrate, and (c, f) unmodified tip and bare substrate. Bias voltage (V_{bias}) was swept from -1.1 V to 1.1 V in 5 ms. Representative transition voltage spectra are shown in insets of d, e, and f. Pink and blue spectra correspond to those with low and high V_{trans} , respectively.”

Comment 7. *Why is it reasonable to compare these data with the data of an 8-mer DNA junction, if there is disorder?*

Response. The definite sequence of the sample DNAs makes the comparison reasonable. Please see the Response to Comment 1 of Reviewer #3.

Comment 8. *the main result is that, basically, stretching does not fully "un-zip" the DNA. This makes comment #1 even more important.*

Response. We agree with the point that the incomplete un-zipping of the structure when stretching is the main finding of the present study. We performed additional AFM experiments to confirm the structure. Please see the Response to Comment 2 of Reviewer #1.

Comment 9. *Why did the authors stop at 10 and 90 base pairs? a much more useful study would be to see the junction restoration effect for junctions with 10,20,30,...90 base pairs. What is the minimal length of DNA that can be used to stabilize the junction this way?*

Response. This question was addressed by the additional experiments on the displacement dependence of the junction restoration behavior (please see the Response to Comment 1 of Reviewer #2). We found that even the 10-base-pair DNA can stabilize the junction at short stretching distances. Given that the

melting temperature of the 10-base-pair DNA is slightly above room temperature, experiments using shorter DNA would cause considerable uncertainties when examining the junction stabilities. We thus refrained from these experiments to further consider the minimal length for the junction restoration effect. The results of the additional experiments were added as Supplementary Note 8. Within this Note, the description particularly relevant to this comment is as follows.

(Supplementary information, Supplementary Note 8, Page 19, Lines 5–9)

“The high correlation at the shorter displacements means that the DNA zipper junctions were regenerated during the two consecutive $G-z$ measurements, demonstrating the restoration capability. It is noteworthy that the restoration behavior was observed even for the junction with 10-mer DNA, though at short displacement. We thus anticipate that dsDNAs stable at room temperature can induce the restoration effect based on the present strategy.”

Comment 10. *In short, while this project is very interesting and exciting, the data presented in this paper is in my opinion too preliminary, and the paper cannot be accepted in its present form. The authors are welcomed to add to this paper and response to my comments.*

Response. Following all the comments of the Reviewers, we now include the results obtained by the additional experiments and calculations. We believe that these additions and revisions address the Reviewers’ concerns.

Additional Changes

Change 1. We added Prof. Noriyuki Kurita of Toyohashi University of Technology as an author of the present work, since he made significant contributions with the additional theoretical calculations. The Author Contributions section was accordingly revised as follows.

(Main text, Page 16, Lines 12 and 13)

“T.H., T.T., and N.K. contributed to theoretical calculations.”

Change 2. Dr. Shintaro Fujii, originally listed as the fifth author, played a prominent role together with the first author in conducting the additional experiments in this revision. We thus changed Dr. Fujii to the second author. The Author Contributions section was revised as follows.

(Main text, Page 16, Lines 11 and 12)

“T.H., S.F., and Y.J. performed single-molecule measurements.”

Change 3. The revised Supporting Information contains many figures. We thus now refer to them using their Note numbers, instead of the Figure numbers, which had been used in the original manuscript. The relevant revisions are as follows.

(Main text, Page 5, Lines 4–7)

“The tunnelling decay constants during and after the plateau (β_1 and β_2 , respectively) were

analysed from each conductance trace, and β_1 and β_2 for 90-mer DNA were determined to be 0.27 and 2.0 \AA^{-1} , respectively (see Supplementary Note 1 for the results of 10-mer DNA and detailed discussion of STM-BJ results).”

(Main text, Page 10, Lines 10–12)

“The dwell length was then compared to the plateau length of the single-molecule junctions (dashed lines in Fig. 3b) of the 90-mer or 10-mer zipper DNA (Supplementary Note 1) to determine whether the molecular junction was successfully formed.”

(Main text, Page 10, Lines 17 and 18)

“The time course of the dwell length was quantitatively analysed using joint probabilities (Supplementary Note 6).”

Reviewer #1 (Remarks to the Author):

The authors have done a considerable amount of additional work, and their conclusions are now much better supported.

Reviewer #2 (Remarks to the Author):

The manuscript "Single-molecule junction spontaneously restored by DNA zipper," by Harashima et al. describes single-molecule junction experiments on DNA where the DNA is placed in the transverse direction to transport (instead of the longitudinal direction) as is typically done in break-junction experiments. The manuscript has vastly improved, but I believe there are still some important questions that should be considered.

First, I am still uncertain of their transport model. As they begin pulling on the DNA, they claim it unzips, and they can still measure conductance over some distance. But how much is unclear. Does the conductance decrease after the first basepair dissociates? It has been previously demonstrated in many cases that single-strand DNA is not conductive, and recently it was shown that conductance can decrease in the longitudinal direction once unzipping begins (Tao). It has also been shown that hydrogen bonded bases or basepairs that are attached to tip and substrate are conductive (Lindsay, Taniguchi, Nagpal, etc.). This work should be placed in context of this other work, noting the differences, and why they may get a higher conductance in this case, and why it may depend on the length to such a degree.

There has also been an improvement of the control experiments, but one important control experiment is still missing. That is with the substrate covered with single-strand, and a bare tip. If the long DNAs are supercoiling, there may be some cases of similar very weak plateaus in these cases. It would be important to do both tapping and IV controls in this case.

Reviewer #3 (Remarks to the Author):

The authors have made substantial corrections and alterations to their manuscript, which include additional experiments and calculations, which pretty much answer most of mine (as well as the other referees) comments. I therefore recommend publication.

List of Revisions

Single-molecule junction spontaneously restored by DNA zipper

Manuscript ID: NCOMMS-19-41904A

Takanori Harashima, Shintaro Fujii, Yuki Jono, Tsuyoshi Terakawa, Noriyuki Kurita, Satoshi Kaneko, Manabu Kiguchi, and Tomoaki Nishino

We are sincerely grateful for the time and effort the reviewer has dedicated again in providing insightful feedback on ways for us to further improve our manuscript. We have incorporated changes that reflect the detailed suggestions you have graciously provided. We hope that our revisions and the responses provided as follows satisfactorily address all the issues and concerns you have noted.

To facilitate your review of our revisions, the following is a point-by-point response to the questions and comments.

Reviewer 1

Comment. *The authors have done a considerable amount of additional work, and their conclusions are now much better supported.*

Response. We respectfully appreciate the Reviewer's previous comments that significantly improved the manuscript.

Reviewer 2

The manuscript "Single-molecule junction spontaneously restored by DNA zipper," by Harashima et al. describes single-molecule junction experiments on DNA where the DNA is placed in the transverse direction to transport (instead of the longitudinal direction) as is typically done in break-junction experiments. The manuscript has vastly improved, but I believe there are still some important questions that should be considered.

Comment. *First, I am still uncertain of their transport model. As they begin pulling on the DNA, they claim it unzips, and they can still measure conductance over some distance. But how much is unclear. Does the conductance decrease after the first basepair dissociates?*

Response. The distance from which the junction conductance was detected can be measured by the length of the plateau in the conductance trace. The plateau length had been included in Supplementary Note 1. We revised this Note to add ample discussion to include the answer to the Reviewer's question and to make the transport model clearer, as follows.

(Supplementary information, Supplementary Note 1, Page S3)

“The fitting for β_1 allowed us to locate the plateau region and evaluate its length, which is equivalent to the distance from which the junction can be stretched before the breakdown, in each conductance trace. Supplementary Fig. 1c shows the histograms for the length of the resulting plateaus, and the median values of the plateau length for 10-mer and 90-mer DNA were determined to be 0.03 and 0.08 nm, respectively. Similar values of the plateau length were reported for the single-molecule junction of double-stranded DNA (dsDNA) in the conventional junction configuration, where the duplex aligned parallel to the axis of the gap between the STM tip and substrate². The plateau length significantly shorter than the molecular length of the DNA has been ascribed to the shear force localized to the terminal base pairs. The mechanical stretch exerted on the DNA by the tip is not evenly distributed along the duplex but is localized at the end base pairs^{3,4}. Consequently, even at the short plateau length, the large shear force arises to mechanically melt the DNA duplex at the termini. The similar plateau length detected in the present and previous works, though different junction configurations were employed, indicates that the breakdown of the present molecular junction that leads to the abrupt decrease in the conductance (see Supplementary Fig. 1a) is also caused by the dissociation of terminal base pair of the duplex. The mechanical stretch in the present experiments was applied directly to the DNA terminal in the direction to rupture the hydrogen bonding in the base pair, which rationalizes the aforementioned mechanism.”

In addition, we corrected the caption of Supplementary Fig. 1c. Although the plateau lengths were determined using the median of the measured values, we described this as the mode by mistake in the caption. The corrected caption is as follows.

(Supplementary information, Caption of Supplementary Fig. 1, Page S2)

“c Plateau length histograms of 10-mer (purple) and 90-mer (blue) DNA. Arrowheads indicate the median values of the plateau length.”

Comment. *It has been previously demonstrated in many cases that single-strand DNA is not conductive, and recently it was shown that conductance can decrease in the longitudinal direction once unzipping begins (Tao). It has also been shown that hydrogen bonded bases or basepairs that are attached to tip and substrate are conductive (Lindsay, Taniguchi, Nagpal, etc.). This work should be placed in context of this other work, noting the differences, and why they may get a higher conductance in this case, and why it may depend on the length to such a degree.*

Response. The literature on the longitudinal electron transport in DNA (Tao) was cited as reference 2 in Supporting Information, and its discussion was added to Supplementary Note 1 quoted in the previous Response.

Regarding the electron transport through DNA bases and hydrogen-bonded base pairs (Lindsay, Taniguchi, Nagpal), we found that the quantitative comparison between previous results

and the present one is difficult. In the previous works, single-stranded DNA was employed (Lindsay, Taniguchi, Nagpal), or DNA bases without the deoxyribose sugar moieties served as samples (Lindsay). The electrical measurements and calculations are distinctly different in the viewpoint of the chemical structures from the present study, where the electron transport via double-stranded DNAs was addressed. Given the notable dependence of single-molecule conductance on a chemical structure of the constituent molecule in the junction, the differences in the conductance between previous and the present studies reflect not only the number of base pairs (single for the previous works and the polymer in the present study) but also the structural variations, the latter of which hinders the quantitative discussion suggested by the Reviewer. Based on this consideration, we cited the recommended literature as reference 28 and 29 (Lindsay), reference 30 (Taniguchi), and reference 27 (Nagpal), and added the relevant discussion to the main text, but we refrain from quantitative arguments about the differences in the single-molecule conductance. These revisions are as follows.

(Main text, Page 5, Lines 11–21)

“On the other hand, the β_1 value is significantly smaller than the typical value for alkanedithiol, but similar to the ones for π -conjugated molecules²⁶, which indicates that the electron transport involves the DNA. The DNA zipper junction transmits electrons in the transverse direction, and it is anticipated that the base pairs, especially those located at the DNA terminal, mediate the electron transport (see Fig. 1a). In this case, the transport properties of the present junction can be compared with those of the single-molecule junction of DNA bases, which have been thoroughly investigated toward the realization of single-molecule sequencing²⁷⁻³⁰. Indeed, the significant reduction in the decay constants as observed for β_1 value in the present experiments was reported for the tunnelling through the DNA base pairs^{31,32}. Further STM-BJ experiments showed that the conductance of the molecular junction reflects the DNA sequence (Supplementary Note 2), which further supports that the electron transport is mediated by the DNA.”

Comment. *There has also been an improvement of the control experiments, but one important control experiment is still missing. That is with the substrate covered with single-strand, and a bare tip. If the long DNAs are supercoiling, there may be some cases of similar very weak plateaus in these cases. It would be important to do both tapping and IV controls in this case.*

Response. Following the comment, we performed the additional experiment: the tapping (STM-BJ) and I - V measurements using the unmodified tip and substrate covered with single-stranded DNA. Both measurements revealed that the single-stranded DNA does not form a single-molecule junction with detectable conductance, and these results indicate that the supercoiled DNA has negligible contributions to the observed conductance. The specific revisions are as follows.

STM-BJ measurements

The results were added to the Supporting Information, and a concise description was included in the main text.

(Main text, from Page 7, Line 22 to Page 8, Line 2)

“However, no state that could be attributed to ssDNA molecular junctions was found in the I - V curves in Fig. 2b and 2c. STM-BJ study was also conducted with the unmodified tip and the ssDNA-modified substrate, and the conductance histograms without notable peaks were obtained (Supplementary Note 5). These results are most probably due to the significantly decreased conductance of ssDNA as compared to that of dsDNA because of base stacking is less ordered in ssDNA³⁶.”

(Supporting Information, Supplementary Note 5, Pages S9 and S10)

“Supplementary Note 5: STM-BJ measurements using unmodified tip and DNA-modified substrate.

STM-BJ measurements were conducted using the unmodified tip and substrate modified with 90-mer single-stranded DNA (ssDNA). The sequence of the ssDNA was the same as that used in the STM-BJ experiments in the main text (Fig. 1). Supplementary Fig. 5a and 5b show the conductance traces and histogram, respectively. Clear plateaus and prominent peaks were found in the traces and histogram, respectively, at integer multiples of $1 G_0$, which originated from Au point contacts between the tip and substrate. No reproducible plateaus were observed in the conductance traces below $1 G_0$ unlike the case where both the tip and substrate were modified with complementary DNAs (Fig. 1). Consequently, no peaks appeared in the conductance histogram below $1 G_0$, though the traces were measured with a wide conductance range by a logarithmic preamplifier. The slight increase in the histogram counts starting from approximately $10^{-2.5} G_0$ were also found with the unmodified tip and unmodified substrate (black line in Supplementary Fig. 5b) and, thus, were attributed to background current noises. These results demonstrate that the ssDNA does not cause the formation of a single-molecule junction having detectable conductance values, most probably due to the significantly decreased conductance of ssDNA as compared to that of dsDNA.”

Supplementary Fig. 5. Typical conductance traces (a) and histogram (b) obtained with the measurements using the unmodified tip and ssDNA-modified substrate. **b** Histogram obtained with the unmodified tip and unmodified substrate was also shown by black line for comparison. The histogram counts were normalized by the values at 1 G₀. Histograms were constructed from 1653 and 2754 traces for the ssDNA-modified and unmodified substrate, respectively. Tip velocity, 10 nm/s; bias voltage, 20 mV.

I–*V* measurements

The data was added to Fig. 2 as panels c and g, and the results were discussed in the main text as follows.

(Main text, Page 7, Lines 12–18)

“Fig. 2a shows the resulting *I*–*V* curves, and Fig. 2b–d presents those obtained in the control experiments, i.e., the measurements with the molecular tip and unmodified surface, with the unmodified tip and ssDNA-modified substrate, and with the unmodified tip and unmodified surface, respectively. The *I*–*V* curves in Fig. 2a clearly exhibited two distributions, i.e., states with high and low conductance. The low-conductance state was common to those found in the control experiments (Fig. 2b–d), indicating that this state stems from the gap devoid of the DNA bridge.”

Fig. 2 Current–voltage (*I*–*V*) and transition voltage spectra. *I*–*V* 2D histograms and V_{trans} histograms obtained with (a, e) molecular tip and modified substrate, (b, f) molecular tip

and bare substrate, **(c, g) unmodified tip and modified substrate**, and **(d, h)** unmodified tip and bare substrate. Bias voltage (V_{bias}) was swept from -1.1 V to 1.1 V in 5 ms. Representative transition voltage spectra are shown in insets of **e–h**. Pink and blue spectra correspond to those with low and high V_{trans} , respectively.

The panels of Fig. 2 were changed as a consequence of adding the results of the control experiment, as quoted above. The labels in the main text were accordingly changed as follows.

(Main text, Page 8, Lines 5 and 6)

“For evaluation of the electronic structure of the molecular junction, the transition voltage (V_{trans}) was estimated using I – V curves of the high-conductance state (Fig. 2**e–h**).”

(Main text, Page 8, Lines 10 and 11)

“The mean value of V_{trans} for the molecular junction of the 90-mer DNA zipper was found to be 0.4 V (Fig. 2**e**).”

Reviewer 3

Comment. *The authors have made substantial corrections and alterations to their manuscript, which include additional experiments and calculations, which pretty much answer most of mine (as well as the other referees) comments. I therefore recommend publication.*

Response. We are most grateful for the Reviewer’s previous comments, which greatly enhanced the quality of the manuscript.

Additional Change

Change 1. As per the request by Editor, we revised the description in the “Data Availability” section as follows.

(Main text, Page 16, Lines 20 and 21)

“The data that support the findings of this study are available from the corresponding author upon reasonable request.”

REVIEWERS' COMMENTS

Reviewer #2 (Remarks to the Author):

The authors have sufficiently addressed my concerns.

List of Revisions

Single-molecule junction spontaneously restored by DNA zipper

Manuscript ID: NCOMMS-19-41904B

Takanori Harashima, Shintaro Fujii, Yuki Jono, Tsuyoshi Terakawa, Noriyuki Kurita, Satoshi Kaneko, Manabu Kiguchi, and Tomoaki Nishino

Reviewer 2

Comment. *The authors have sufficiently addressed my concerns.*

Response. We are most grateful for the Reviewer's previous comments, which greatly enhanced the quality of the manuscript.